# Theoretical studies on anhydride dynamic covalent bond exchange mechanisms

Xinglong Zhang [1,2] ✉, Qiubo Chen[1], Nannan Li[1], Michael B. Sullivan [1] & Jianwei Zheng [1] ✉

Anhydrides have been utilised to synthesize covalent adaptable networks (CANs), enabling the recycling of thermosets and immiscible plastic wastes due to their dynamic bond exchange properties. However, the exact bond exchange mechanism involving anhydrides remains unclear. Herein, we identify the bond exchange mechanism behind anhydride-based CANs through a comprehensive study of the anhydride bond exchange mechanism under uncatalysed and acid-catalysed reaction conditions using density functional theory (DFT) methods. Careful conformational samplings of structures and transition states have been performed to ensure accurate determination of the key rate-determining step. The bond exchange barrier increases from 44.1 kcal mol$^{-1}$ at 25 °C to 52.8 kcal mol$^{-1}$ at 200 °C for the uncatalysed route and from 25.9 kcal mol$^{-1}$ to 33.0 kcal mol$^{-1}$ over the same temperature range for acid-catalysed route. Our results implicate that anhydrides can be good dynamic covalent bond linker candidates in CANs for high temperature applications and acid catalysts may be used as mediators to assist low-temperature reprocessing and recycling.

Thermosetting polymers, an important class of high-performance polymers, are used in many industrial applications[1]. However, due to the presence of permanent covalent crosslinks, these materials are impossible to reprocess and therefore difficult to recycle. Recently, the emergence of covalent adaptable networks (CANs), which are permanent networks containing dynamic linkages to impart reprocessability at high temperature, has opened a new avenue for tackling this issue[2–5]. In addition, dynamic covalent bonds have been reported as universal dynamic crosslinkers (UDCs) to make immiscible plastics miscible and reprocessable[6], suggesting an alternative way to recycle mixed plastic wastes. Furthermore, dynamic covalent bonds have also been incorporated into thermoplastic polymers to adjust the reprocessing temperatures to reduce the energy consumption and potential degradations[7–9].

Reprocessing temperature is one of the key properties for CAN materials and is closely related to the topology freezing transition temperature ($T_v$). $T_v$, proposed by Leibler et al.[5,10] represents the upper service temperature and lower reprocessing temperature, and is defined as the point at which the viscosity becomes higher than $10^{12}$ Pa s. $T_v$ can be measured by using a dilatometry test or stress-relaxation test by rheology[5] or dynamic mechanical analyser[11]. However, external forces and many other experimental factors may strongly affect the results. For example, fluorescent probes need to be incorporated into CANs in aggregation-induced-emission method[12]. As the rheological properties of CAN materials are closely related to the kinetics of dynamic covalent bond exchange[13–15], understanding the dynamic bond exchange mechanism and obtaining the

bond exchange activation energy could pave the way for estimating reprocessing temperatures and thereby designing CANs of desirable properties.

Various types of dynamic covalent bonds in CANs have been reported in the literature[2]. Although bond exchanges are observed in these CANs, detailed studies on bond exchange mechanisms are limited. Density functional theory (DFT) simulations have proven to be a powerful tool in exploring the reaction and bond exchange mechanisms. For bond exchange mechanisms in aromatic and aliphatic disulfide compounds, DFT simulations showed that the [2 + 1] radical-mediated mechanism, rather than the [2 + 2] metathesis mechanism, is the one responsible for the self-healing process of vitrimers with disulfide dynamic linkages (Scheme 1i)[16]. For exchange reactions between boronic esters, DFT simulations show that both metathesis and nucleophile-mediated pathways are possible, although an accelerated rate of exchange in the presence of a nucleophile is observed (Scheme 1ii)[14]. In this case, the nucleophile very likely acts as a catalyst for the bond exchange. As for the metathesis of acetal linkages, DFT simulations corroborated that acetal linkages substituted with aromatic rings may proceed via a carbocation intermediate in a dissociative exchange mechanism, facilitated by acid catalysis (Scheme 1iii)[17]. The presence of the strongly electron-withdrawing fluorine atoms and the basicity of tertiary amines that activate the thia-Michael exchange were also explored via a DFT mechanism study, where an internal amine catalyst is proposed to facilitate the bond exchange (Scheme 1iv)[18]. We note that most of these bond exchange mechanisms require some kind of catalyst. Theoretical studies on

[1]Institute of High Performance Computing, Agency for Science, Technology and Research (A*STAR), Connexis, Singapore, Singapore. [2]Department of Chemistry, The Chinese University of Hong Kong, Shatin, New Territories, Hong Kong, China. ✉e-mail: xinglong.zhang@cuhk.edu.hk; zhengjw@a-star.edu.sg

**i) Disulfide-based covalent bond exchange**

[2+2] metathesis

[2+1] radical mediated stepwise mechanism

**ii) Boronic esters-based covalent bond exchange**

**iii) Acetal-based covalent bond exchange**

**iv) Thia-Michael covalent bond exchange**

**v) Dynamic covalent exchange between methacrylic anhydride and 4-pentenoic anhydride resulting in a mixed anhydride**

methacrylic anhydride    4-pentenoic anhydride

**Scheme. 1 | Examples of covalent bond exchange systems and their potential mechanisms for bond exchange. i** Disulfide-based covalent bond exchange. **ii** Boronic acid-based covalent bond exchange. **iii** Acetal-based covalent bond exchange. **iv** Thia-Michael covalent bond exchange. **v** Dynamic covalent bond exchange between methacrylic anhydride and 4-pentanoic anhydride (present work).

anhydride covalent bond exchange reactions, on the other hand, have not been reported.

Bond exchange reactions involving anhydride have recently been utilised to synthesize CANs. Lawton et al.[19] reported the development of soft-shape memory polymers using polyanhydride-based elastomers as the permanent elastic phase. They demonstrated that dynamic anhydride exchange reactions at a temperature of 90 °C in 9 min or at 50 °C for 4 h allow almost complete permanent shape reconfiguration in the solid state[19]. They proposed a bond exchange mechanism involving a 4-member ring transition state without any detailed study, either experimentally or theoretically. Tillman et al. examined experimentally the dynamic anhydride exchange reactions in both model compounds (Scheme 1v) and poly(thioether anhydrides) networks in terms of the equilibrium constant, the activation energy, and the rate of exchange without detailed reaction pathways[20]. They demonstrated that the crosslinked poly(thioether anhydrides) exhibited self-healing and facile recycling behaviour, which allows for the complete recovery of damage within 4 h at 90 °C. It was also demonstrated that recycling under compression moulding at 90 °C could be achieved, indicating that bond exchange mechanisms may be operative under force. In 2023, Clarke et al. reported using anhydrides as UDCs to make immiscible plastics miscible and reprocessable because of anhydride exchange reactions[6]. They incorporated crosslinkers containing anhydride dynamic covalent bonds into pure polyolefins and an immiscible mixture of polyolefins with biodegradable polymers such as poly-3-hydroxybutyrate (P3HB) and poly-L-lactic acid (PLLA) through melt extrusion at 180 °C, even when colorant and other additives exist. They observed enhanced mechanical properties that did not reduce after reprocessing. They

attributed the good mechanical properties to stable anhydride dynamic bonds. Santefort et al.[21] prepared copolymers of methacrylic anhydride (MAA) with methyl methacrylate (MMA) or butyl acrylate (BA). Interchain crosslinking due to bond exchange between anhydrides significantly increases the glass transition temperature to ~123 °C for 50/50 MAA/BA copolymers, which is much higher than 26 °C calculated using Fox equation[22]. They also observed that recycling at 130 °C for 2 h under 152 MPa pressure will not sacrifice the mechanical properties.

It is interesting to note that the reprocessing temperatures differ among all of these reported CANs, spanning from 50 °C to 180 °C[6,19–21]. The detailed uncatalysed anhydride exchange reaction mechanism using either experimental or computational investigations has not yet been reported. On the other hand, the acid-catalysed anhydride exchange mechanism has been studied experimentally by Mironov and Zharkov, whose earlier works showed that anhydride bond exchange can occur readily under acid-catalysis[20]. In 2021, Clarke et al. reported an energy barrier of 48.0 kcal mol$^{-1}$, which corresponds to the Gibbs free-energy barrier of ~36 kcal mol$^{-1}$ at 200 °C (Supplementary Information Section II.7) for acid/anhydride bond exchange in CANs prepared from a random copolymer of n-butyl acrylate and acrylic acid, however, without providing direct evidence on the mechanism[23]. The mechanism behind the reprocessability of anhydride-based CANs remains an open question.

Herein, we report a detailed study of the uncatalysed and acid-catalysed anhydride exchange mechanisms using DFT simulations. Using aliphatic anhydrides as model compounds, we explored all possible routes for anhydride bond exchange reactions. Although the real polymer environment is more complex, a mechanistic study using model compounds is

**Scheme. 2 | Mechanistic possibilities for dynamic covalent bond exchange between methacrylic anhydride (MAA) and 4-pentenoic anhydride (PNA). i** Proposed reaction mechanism via a concerted mechanism (6- or 4m ring) or b stepwise mechanism. **ii** Alternative mechanism involving the stepwise C–O bond breaking, forming a quaternary carbon centre intermediate via a 6-membered transition states or b) 4-membered transition states.

necessary. Such studies lay the foundation for further study. We found the bond exchange barrier for uncatalysed route is as high as 44.1–52.8 kcal mol$^{-1}$, whereas the barrier for the acid-catalysed route can be lowered to 25.9–33.0 kcal mol$^{-1}$, suggesting that anhydride dynamic covalent bonds could be a good candidate for high temperature applications, and reprocessing or recycling can be achieved with an acid catalyst under reasonable reaction temperatures.

## Results

### Mechanism of dynamic covalent bond exchange between MAA and 4-pentenoic anhydride (PNA)

In our benchmarking studies (Supplementary Information Section II.1), we note that Gibbs free energies are comparable (within 2–4 kcal mol$^{-1}$) between gas-phase and solvent-phase optimised geometries, with larger RMSD values (0.200–0.207 Å) correlating with slightly higher energy differences; Pople basis sets with diffuse functions yielded Gibbs free energies ~2–3 kcal mol$^{-1}$ lower than Karlsruhe basis sets. Single-point energy corrections using SMD(CHCl$_3$)-DLPNO-CCSD(T)/CBS versus SMD(CHCl$_3$)-M06-2X/def2-TZVP showed nearly identical Gibbs free energies for intermediates and differences within 2 kcal mol$^{-1}$ for transition states, indicating both methods are suitable. Due to the balanced description provided by Karlsruhe basis sets and the high accuracy of DLPNO-CCSD(T)/CBS, SMD(CHCl$_3$)-DLPNO-CCSD(T)/CBS//M06-2X/def2-SVP was chosen for all subsequent

studies, and the Gibbs free energy profiles produced using this method are used for discussion throughout.

The possible mechanisms for the covalent bond exchange between MAA and PNA, in the absence of a catalyst, are shown in Scheme 2. Firstly, a concerted mechanism may be possible, allowing for a single-step formation of mixed anhydride (Scheme 2ia). The transition state (TS) structures, either via 6-membered (**TS_concerted_6m**, Supplementary Fig. 3) or 4-membered[19] (**TS_concerted_4m**, Supplementary Fig. 3) ring, were both found to have a barrier height of 59.3 kcal mol$^{-1}$ (Supplementary Information Section II.2), which is higher than TS barriers via other mechanisms (vide infra) and is thus ruled out.

A stepwise mechanism was subsequently considered. In this mechanism, the oxygen atom of one anhydride acts as a nucleophile attacking the carbonyl group of the other anhydride partner, forming an ion-pair intermediate consisting of the oxonium cation and the carboxylate anion (Scheme ib). Subsequent attack of the oxonium cation by the carboxylate anion forms the final product of mixed anhydride. The TS structure for the first step of this mechanism could not be located. A relaxed PES scan along the anhydride oxygen atom of MAA and the carbonyl carbon atom of PNA was performed, indicating a barrier of potentially as high as 60 kcal mol$^{-1}$ (Supplementary Fig. 4). This mechanism is disfavoured, possibly due to the formation of charge-separated ion-pair intermediates. We performed a similar scan with an implicit solvent model and obtained similar results, suggesting that this mechanism is unlikely.

**Scheme. 3 | Anhydride bond exchange via stepwise C–O bond breaking, forming a quaternary carbon centre intermediate via 6-membered transition states.** Pathways involve breaking C–O bond in (**a**) methacrylic anhydride (MAA) and (**b**) 4-pentenoic anhydride (PNA).

a) Anhydride exchange breaking the C–O bond in methacrylic anhydride (MAA)

b) Anhydride exchange breaking the C–O bond in 4-pentenoic anhydride (PNA)

From the relaxed PES scan result (Supplementary Information Section II.3), we hypothesize that the anhydride C–O σ-bond on one species may break and add across the C=O bond of another anhydride species, forming an intermediate resembling structure 4 in Supplementary Fig. 4, with a quaternary carbon centre. We therefore proposed an alternative mechanism involving either a 4-membered or a 6-membered transition state as shown in Scheme 2ii.

The 4-membered TS was initially located and used for benchmarking studies with the Gibbs free-energy profile, without any conformational sampling of each structure, shown in Supplementary Fig. 1. The energy profile suggests that the first step has the highest barrier, with **TS1-4m** at 58.8 kcal mol$^{-1}$ relative to the sum of the separated reactants. In Supplementary Fig. 1, the MAA C–O σ-bond is broken in the 4-membered TS structure, **TS1-4m**. We considered the second possibility where the PNA σ-bond is broken in a 4-membered TS structure (**TS1'-4m**) and the Gibbs free-energy profile is shown in Supplementary Fig. 5. The first step has a barrier of 57.4 kcal mol$^{-1}$ and is overall the rate-determining step. Similar consideration of the mechanism via a 6-membered TS structure suggests again that step 1 is the rate-determining step, at 51.0 kcal mol$^{-1}$ (Supplementary Fig. 6). As the pathway via 6-membered TSs has a much lower barrier (>6 kcal mol$^{-1}$) than the pathway via 4-membered TSs, we subsequently considered reaction pathways involving 6-membered TSs.

**Conformational effects on the reaction potential energy surface**

In our attempt to locate the TSs for different pathways, we tried to keep the orientations of the side groups consistent for a fair comparison of the TS barriers. To improve upon our picture of the PESs, we performed detailed conformational analyses for each species. Grimme's *CREST* program[24,25], running at the GFN2-xTB[26–28] extended semiempirical tight-binding level of theory, was used to search for all conformers at each step. For the search for TS conformers, key bond distances were constrained before subjecting the structure to the *CREST* program. The resulting lowest 20 energy structures on the xTB PES were further optimised at the DFT level to obtain the final results.

Scheme 3 shows all the possibilities for the reaction mechanism for the anhydride bond exchange between two distinct anhydrides via 6-membered TSs. In step 1, either anhydride may break its C–O σ-bond to add across the C=O π-bond of the other anhydride, forming an intermediate with quaternary carbon (INT2 and INT2', differing in the alkyl group at the quaternary carbon centre). In step 2, three possibilities exist for the anhydride formation (each of the three carboxylate groups can combine with another): one will revert to the starting materials, while the other two will form the mixed anhydride product.

Upon full conformational sampling of all species, the resulting Gibbs free-energy profiles are shown in Fig. 1; the DFT-optimised structures of the

**Fig. 1 | Gibbs free-energy profile for the dynamic covalent bond exchange via 6-membered TSs.** Gibbs free energies were computed at SMD(CHCl₃)-DLPNO-CCSD(T)/CBS//M06-2X/def2-SVP level of theory at 25 °C, for (**a**) Pathway I shown in Scheme 3a and (**b**) Pathway II shown in Scheme 3b after full conformational sampling.

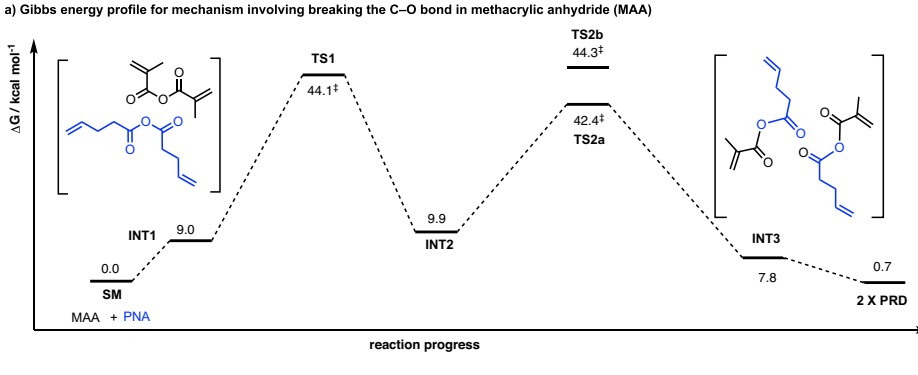

a) Gibbs energy profile for mechanism involving breaking the C–O bond in methacrylic anhydride (MAA)

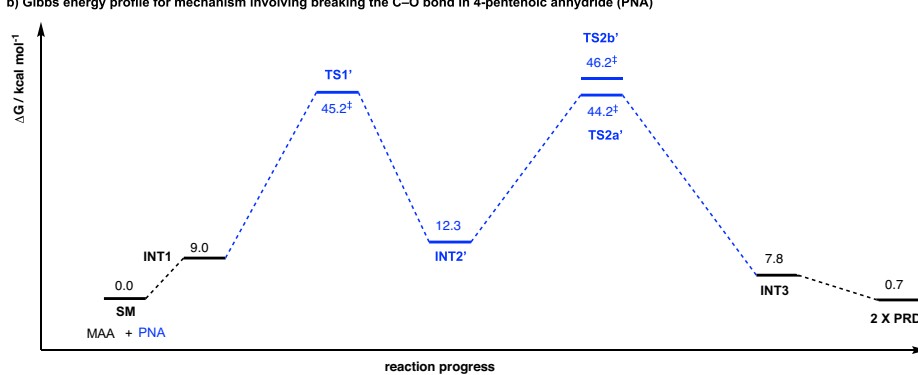

b) Gibbs energy profile for mechanism involving breaking the C–O bond in 4-pentenoic anhydride (PNA)

lowest energy transition state structures are shown in Fig. 2. Note that **INT1** is the reactant complex consisting of one molecule of MAA and one molecule of PNA. The reactant complexes **INT1-4m**, **INT1'-4m**, and **INT1-6m** are different conformers of **INT1**. As these conformers will be in rapid thermal equilibrium, the lowest energy reactant complex will be the same for all pathways (**INT1** in this case), whether the reaction mechanism occurs via a 4-membered or a 6-membered TS structure. **INT3** is the most stable product complex, consisting of two molecules of mixed anhydride, which is the same whether from 4-membered or 6-membered TS pathways.

Inspecting the Gibbs free-energy profiles in Fig. 1, we see that in the first step, breaking the C–O bond in MAA (via **TS1**, at 44.1 kcal mol⁻¹) is slightly more favoured than breaking the C–O bond in PNA (via **TS1'**, at 45.2 kcal mol⁻¹) by 1.1 kcal mol⁻¹. Although this energy difference is small, as we have considered 20 conformers for each TS and are predicting the difference in the activation barriers (ΔΔG), we may conclude that **TS1** would be more favoured than **TS1'** (this translates to a kinetic preference for **TS1** by a factor of 6.4:1 over **TS1'** using simple transition state theory, without Boltzmann weighting of all conformers). We further note that with Boltzmann weighting (Supplementary Information Section II.11), the PES does not change substantially, and the first step remains as the rate-determining step. Bond dissociation enthalpy (BDE) calculations (Supplementary Information Section II.6) suggest that the C–O bond strengths in MAA and PNA are similar, with a BDE value of 99.8 kcal mol⁻¹ for MAA and 99.3 kcal mol⁻¹ for PNA. The difference in TS barriers also depends on non-covalent interactions of the side groups in the TS structures on top of the C–O bond strengths. Additionally, a diradical mechanism involving the homolytic cleavage of C–O bond in either MAA or PNA was ruled out as these have barriers >80 kcal mol⁻¹ (Supplementary Information Section II.6).

In the second step following **TS1** (Pathway I, Scheme 3a), **INT2** can undergo either Pathway IA via **TS2a** or Pathway IB via **TS2b** to give the mixed anhydride product. **TS2a**, which involves the migration of the pentenoyl group (Scheme 3 and Fig. 2), has a barrier of 42.4 kcal mol⁻¹. This is lower than **TS1** by 1.7 kcal mol⁻¹. On the other hand, **TS2b**, which involves the migration of the methacryloyl group (Scheme 3 and Fig. 2), has a barrier

of 44.3 kcal mol⁻¹. This is almost isoenergetic to **TS1** barrier. Overall, the reaction will go through **TS1** followed by **TS2a** to give mixed anhydride products, with **TS1** as the rate-determining step. Similarly, in the second step following **TS1'** (Pathway II, Scheme 3b), **INT2'** can undergo either Pathway IIA via **TS2a'** or Pathway IIB via **TS2b'** to give the mixed anhydride product. **TS2a'**, involving the migration of the pentenoyl group (Scheme 3 and Fig. 2), is lower in barrier than **TS1'** (by 1.0 kcal mol⁻¹), whereas **TS2b'**, involving the migration of the methacryloyl group (Scheme 3 and Fig. 2), is higher (by 1.0 kcal mol⁻¹). Nevertheless, for this pathway following **TS1'**, **TS2a'** will be kinetically preferred to **TS2b'**, making step 1 the rate-determining step, as in previous cases.

Comparing the Gibbs free-energy profiles with full conformational analyses (Fig. 2) and the Gibbs free-energy profile without consideration of conformations (Supplementary Fig. 5), we see that 1) the barrier height can be lowered by up to 9.3 kcal mol⁻¹ (**TS1-6m** at 53.4 kcal mol⁻¹, Supplementary Fig. 5 to **TS1** at 44.1 kcal mol⁻¹, Fig. 2); and 2) the lowest barrier TS changed in that without conformational considerations, the first step TS breaking the C–O bond in PNA (**TS1'-6m**, Supplementary Fig. 5) is favoured over the TS breaking the C–O bond in MAA (**TS1-6m**, Supplementary Fig. 5), whereas with conformational analyses, the first step breaking the C–O bond in MAA (**TS1**, Fig. 2a) is favoured over the TS breaking the C–O bond in PNA (**TS1'**, Fig. 2a). Thus, conformational considerations may need to be taken when considering e.g., competing selectivities in different reaction mechanisms and/or selectivity-determining steps in computational reaction mechanisms studies[29–39].

With these full conformational considerations, we conclude that step 1 (**TS1**) is the overall rate-determining step. This TS has a barrier height of 44.1 kcal mol⁻¹, which is rather high and cannot occur under room temperature. This barrier is also much higher than the one measured experimentally[20] using ¹³C NMR in CDCl₃, which gives a barrier of about 26.2 kcal mol⁻¹ (see Supplementary Information Section II.7).

## Acid-catalysed anhydride bond exchanges

We note that in the presence of free acids, the reaction barriers may be drastically reduced[20], thus allowing the reaction to occur at much milder

**Fig. 2 | DFT-optimised structures of key transition states for the dynamic covalent bond exchange between methacrylic anhydride (MAA) and 4-pentenoic anhydride (PNA) via 6-membered TSs.** Gibbs free energies were computed at SMD(CHCl₃)-DLPNO-CCSD(T)/CBS// M06-2X/def2-SVP level of theory at 25 °C.

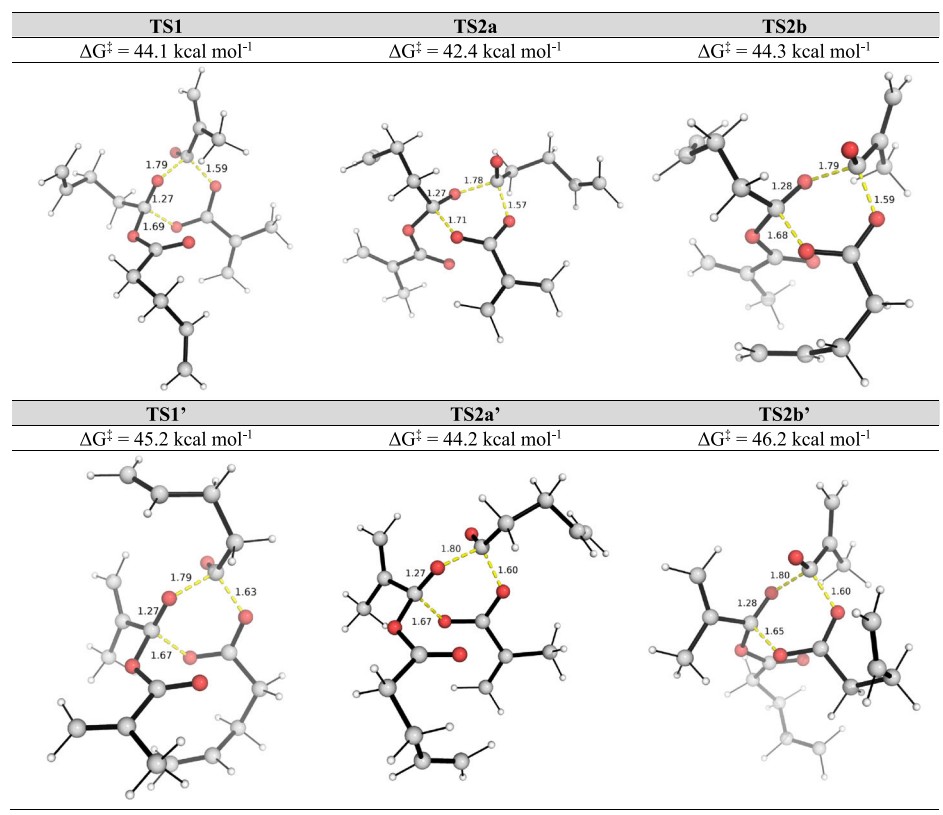

| TS1 | TS2a | TS2b |
|---|---|---|
| $\Delta G^{\ddagger} = 44.1$ kcal mol$^{-1}$ | $\Delta G^{\ddagger} = 42.4$ kcal mol$^{-1}$ | $\Delta G^{\ddagger} = 44.3$ kcal mol$^{-1}$ |

| TS1' | TS2a' | TS2b' |
|---|---|---|
| $\Delta G^{\ddagger} = 45.2$ kcal mol$^{-1}$ | $\Delta G^{\ddagger} = 44.2$ kcal mol$^{-1}$ | $\Delta G^{\ddagger} = 46.2$ kcal mol$^{-1}$ |

temperatures. Computationally, the protonation of MAA by a proton is more favourable than the protonation of PNA, by 1.4 kcal mol$^{-1}$. A plausible mechanism and the computed Gibbs free-energy profile for the anhydride exchange between MAA and PNA under acid catalysis are shown in Fig. 3. In this mechanism, an 8-membered cycloaddition reaction occurs to give the mixed anhydride and acetic acid and an oxonium intermediate. In the subsequent step, the acetic acid attacks the oxonium to give the protonated mixed anhydride, with negligible barrier. A computed reaction barrier of 25.9 kcal mol$^{-1}$ (**TS_acid**, Fig. 3) was found after computational sampling of the transition states. This barrier is very close to the experimental barrier of 26.2 kcal mol$^{-1}$ measured experimentally[20] using $^{13}$C NMR in CDCl₃ (which may contain residual acids), indicating that an acid-catalysed mechanism is likely to be operative for anhydride bond exchange.

## Electronic structure analyses of key TSs

To gain a better understanding of the electronic structure description of the transition states of both catalysed and uncatalysed pathways, we performed natural bond orbital charge analyses and frontier molecular orbital calculations (HOMO/LUMO plots) for key transition states (see Supplementary Information Section II.12). The NBO analyses revealed consistent electron distributions across all uncatalysed transition states (Fig. 4 and Supplementary Fig. 15): O atoms exhibit negative NBO charges ranging from −0.587 to −0.732 a.u., while the adjacent carbon atoms carry positive charges between 0.873 and 0.983 a.u. This supports a concerted mechanism involving nucleophilic attack of oxygen on electrophilic carbon. In contrast, in the acid-catalysed **TS1_acid**, one O atom shows a much less negative charge (−0.369 a.u., Fig. 4 and Supplementary Fig. 15), consistent with the proposed formation of an acylium ion-like intermediate (R–C≡O⁺).

Frontier orbital analysis further revealed that the HOMO is largely localized on peripheral π bonds (C=C), and the LUMO is distributed on carboxylate π* orbitals (Fig. 4 and Supplementary Fig. 15). This may suggest that the key electronic reactivity is driven by groups outside the 6-membered ring scaffold. In addition, activation-strain analysis[40–42] indicates that the major barrier to bond exchange arises from distortion of the reactant

fragments as they reorganise into the 6-membered TS structures, rather than favourable interaction energies (Supplementary Table 6). The acid-catalysed TS may benefit from enhanced interaction stabilisation, possibly due to charge-dipole interactions, leading to much lower barriers.

## Discussion

CANs and vitrimers are materials with dynamic covalent bonds, allowing for reprocessability and recyclability through bond exchange. The bond exchange temperature is closely related to the bond exchange barriers. Ideally, this temperature should be higher than the polymer's application temperature to preserve its required properties and maintain structural integrity, yet not excessively high so as to minimize the energy consumption during reprocessing or recycling.

Among the dynamic covalent bonds explored, anhydrides exhibit relatively high bond exchange barriers of ~44 kcal mol$^{-1}$ for aliphatic anhydrides, in the absence of any acid catalysis, as indicated by our calculations. While these elevated bond exchange barriers suggest that anhydrides could be promising candidates as dynamic covalent crosslinkers for high-temperature applications, the barriers are too high for reprocessing and recycling. However, all experimentally synthesized CANs having anhydrides reported so far can be reprocessed without adding any catalysts at temperatures ranging from 50 °C to 180 °C as demonstrated in various studies[6,19–21]. Therefore, alternative mechanisms exist for the bond exchange in these anhydride systems.

Our calculations on acid-catalysed bond exchange mechanisms indicate that these barriers can be reduced to 25.9 kcal mol$^{-1}$ at 25 °C, which agrees very well with the reported experimental barrier of 26.2 kcal mol$^{-1}$ in CDCl₃, suggesting that the acid residues could facilitate reprocessability or recyclability. Tillman and co-workers observed the bond exchange barrier of ~17.2 kcal mol$^{-1}$ in poly(thioether anhydrides), which is much smaller than that in small model systems for uncatalysed bond exchange of anhydrides. They attribute the low bond exchange barrier to carboxylic acid residues[20]. Similarly, Clarke and co-workers also reported carboxylic acid-anhydride bond exchange in copolymers of n-butyl acrylate or methyl acrylate with

**Fig. 3 | Mechanism of anhydride covalent bond exchange between methacrylic anhydride and 4-pentenoic anhydride under acid catalysis and the associated Gibbs free-energy profile.** Gibbs free energies were computed at SMD(CHCl$_3$)-DLPNO-CCSD(T)/CBS//M06-2X/def2-SVP level of theory at 25 °C.

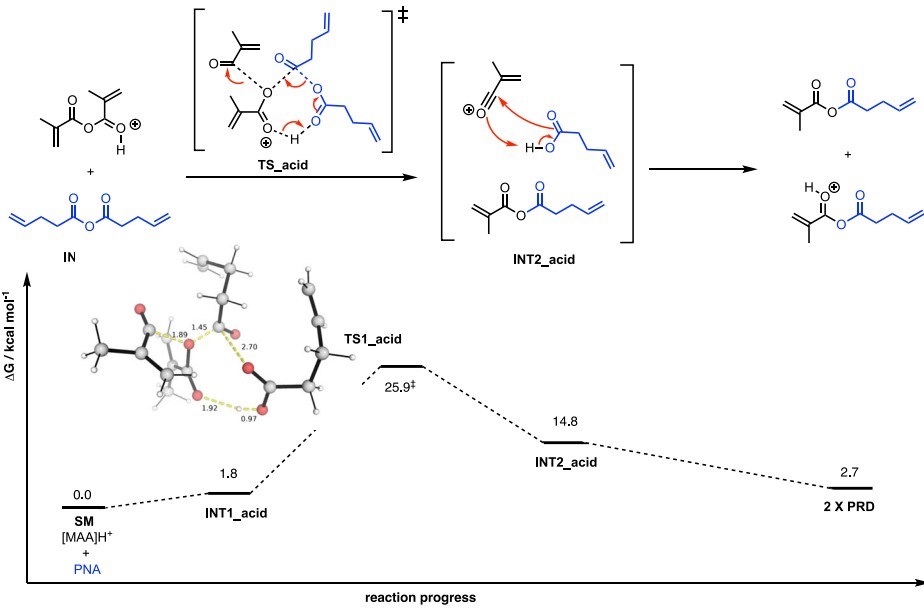

acrylic acid[23]. The existence of residue carboxylic acids is due to their synthesis route, in which anhydrides were produced by the condensation of carboxylic acids. The importance of acids in initialising and/or speeding up catalytic transformations has also been demonstrated in a study where Brønsted acids such as H$_2$SO$_4$ can catalyse the self-initiation of polymerisation in vinyl monomers, in the absence of external initiators[43]. The high Gibbs free-energy barrier of ~36 kcal mol$^{-1}$ at 200 °C (Supplementary Information Section II.7), obtained for acid/anhydride bond exchange in CANs prepared from random copolymer of n-butyl acrylate and acrylic acid[23], agrees well with the predicted barrier of acid-catalysed mechanism of 33.0 kcal mol$^{-1}$ at 200 °C (Supplementary Information Section II.9). The higher barrier could also result from the low concentration of free carboxylic acids and thus lowered probability of encounter between the acids and anhydride groups. We note that our simplified mechanistic modelling treated the two reactants as independent molecules, which may overestimate the entropic penalty for bond formation. In the condensed polymer matrix, translational and rotational degrees of freedom of reacting groups are already restricted, thus reducing the effective entropic loss upon bond formation, and the actual free energy barrier for such processes may be lower than predicted in our model. It is therefore reasonable, based on our computed mechanistic pathways, to conclude that acid-catalysed anhydrides bond exchange reactions are important in the reprocessing of anhydride-based CANs. In a way, anhydride dynamic crosslinkers could be effectively utilised for low to medium temperature applications such as self-healing, shape memory, and composites[44] when incorporating weak or mild acid neighbouring groups.

## Conclusions

We herein report the full reaction mechanism for the uncatalysed anhydride bond exchange and acid-catalysed anhydride bond exchange between two different anhydride monomers, namely methacrylic anhydride and PNA, using DFT methods. Despite a simplified model, our mechanistic elucidations provide valuable insights into the critical role of acids in otherwise challenging anhydride bond exchange processes. A stepwise mechanism involving the breaking of the anhydride C–O bond, which adds across the C=O bond of another anhydride to form a quaternary carbon intermediate, is proposed. This intermediate subsequently undergoes acyl group migration via a 6-membered transition state, leading to the formation of mixed anhydride products. Compared to the uncatalysed bond exchange reaction, the acid-catalysed anhydride bond exchange reaction gives much lower bond exchange barriers and could be the operative mechanism underlying

the reprocessability and recyclability of anhydride-based CANs. Future work will focus on understanding how various polymer environments or neighbouring groups influence the bond exchange behaviour of anhydrides using multi-scale modelling as well as molecular dynamics.

## Computational methods

DFT calculations were performed with Gaussian[45] and ORCA[46–48] software using our jobs submission automation toolkit, CHEMSMART[49]. Initially, for benchmarking, geometry and transition state (TS) optimisations were performed with M06-2×[50] functional with either Karlsruhe basis sets def2-SVP[51,52] or Pople basis sets 6–31 + G(d)[53] in either gas or SMD[54] implicit solvation model. M06-2X functional was chosen as it has been employed in the studies of a range of organic systems, including radical chemistry[55–59] and asymmetric catalysis[29–39], as well as mechanism modelling[4,60] and electronic structure elucidation[61] in materials with good accuracy.

Potential energy surface (PES) minima and TSs were ascertained by vibrational frequencies analysis at the same level of theory as the optimisations, to show that the minima and TSs contain exactly zero and one imaginary frequency, respectively. Single point calculations were performed on the respective DFT-optimised structures, giving four different levels of theory used for benchmarking: SMD(CHCl$_3$)-M06-2X/def2-TZVP//M06-2X/def-SVP, SMD(CHCl$_3$)-M06-2X/6-311 + + G(d,p)//M06-2X/6-31 + G(d), SMD(CHCl$_3$)-M06-2X/def2-TZVP//SMD(CHCl$_3$)-M06-2X/def-SVP and SMD(CHCl$_3$)-DLPNO-CCSD(T)[62,63]/CBS(complete basis set, Extrapolate(cc-pVDZ, cc-pVTZ))//M06-2X/def2-SVP (See Supplementary Information for more details). Solvent effect for chloroform was modelled using SMD implicit solvation to account for the effect of chloroform solvent that was used in the experimental measurement of anhydride exchange rates[20] using $^{13}$C NMR in CDCl$_3$. Where needed, intrinsic reaction coordinate[64,65] analysis was carried out to connect the appropriate reactant/product states passing through a given transition state.

Gibbs free energies were evaluated at different temperatures of interest (See Supplementary Information Section II.9), using Grimme's scheme of quasi-RRHO treatment of vibrational entropies[66], using the GoodVibes code[67]. Vibrational entropies of frequencies below 100 cm$^{-1}$ were obtained according to a free rotor description, using a smooth damping function to interpolate between the two limiting descriptions. The free energies reported in Gaussian from gas-phase optimisations were further corrected using a standard concentration of 1 mol/L[68–71], which were used in solvation calculations, instead of the gas-phase 1 atm used by default in the Gaussian program. For production studies, the SMD(CHCl$_3$)-DLPNO-CCSD(T)/

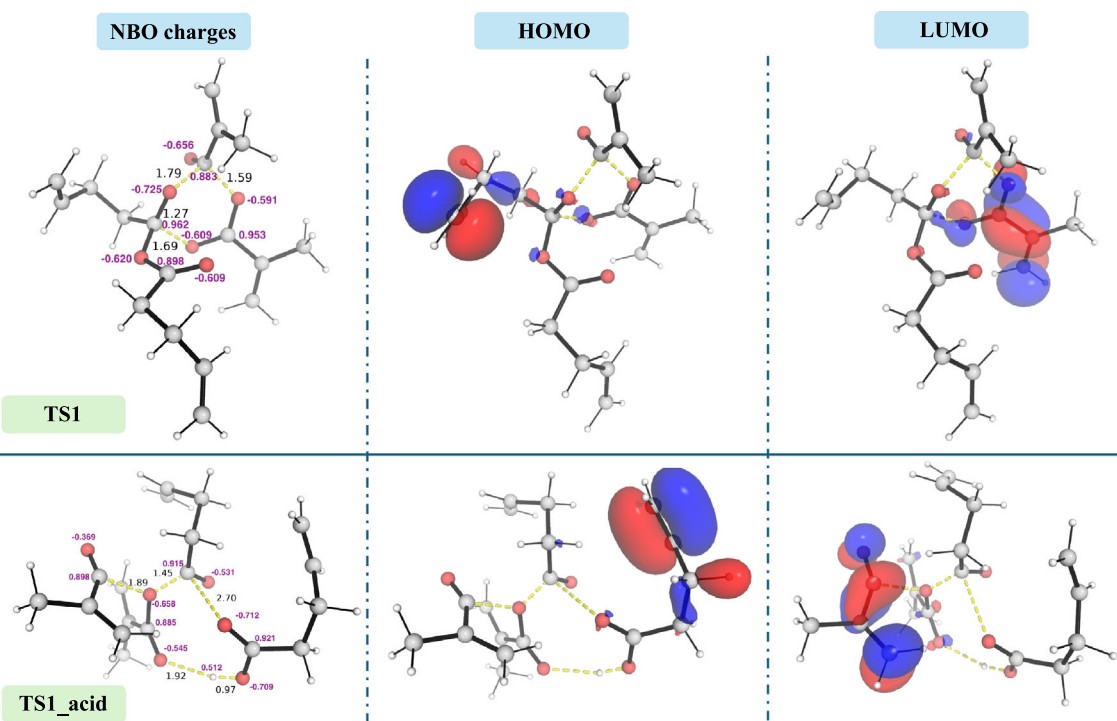

**Fig. 4 | Electronic structure analysis of the lowest energy TSs for both uncatalysed (TS1) and acid-catalysed (TS1_acid) reaction pathways.** Natural bonding orbital (NBO) charges for key atoms and frontier molecular orbital (FMO) plots, including HOMO and LUMO plots for **TS1** and **TS1_acid** are shown.

CBS//M06-2X/def2-SVP level of theory was adopted, and their computed Gibbs free energies are used for discussion throughout. DFT structures and molecular orbitals are visualised using PyMOL[72] software, which .pse files were created using the CHEMSMART toolkit.

## Data availability
DFT-optimised structures from this study in *.xyz* format have been deposited as Supplementary Data 1 together with the Supplementary Information. These have also been uploaded to and are freely available at https://zenodo.org/records/16625442 (DOI: 10.5281/zenodo.16625442).

## Code availability
Automation toolkit codes for quantum chemical calculations are freely available at https://github.com/xinglong-zhang/chemsmart and codes used for thermochemistry analysis in this work are freely available at https://github.com/patonlab/GoodVibes.

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

## Acknowledgements

This project is supported by RIE2025 Manufacturing, Trade and Connectivity (MTC) Programmatic Fund (M22K9b0049) administered by A*STAR.

## Author contributions
J.W.Z. obtained funding and conceptualised the project. X.Z. designed and performed the computational studies with assistance from Q.C., N.L. and M.B.S. X.Z. analysed the results. J.W.Z. provided feedback and oversaw the project. X.Z. and J.W.Z. wrote the manuscript. All authors contributed to the discussion of the results and approved the final manuscript.

## Competing interests
The authors declare no competing interests.
