## [Transparent Peer Review file · Communications Chemistry]

Theoretical studies on anhydride dynamic covalent bond exchange mechanisms

Corresponding Author: Professor Xinglong Zhang

Version 0:

Reviewer comments:

Reviewer #1

(Remarks to the Author)

This manuscript presents a theoretical investigation of dynamic covalent bond exchange in anhydrides, comparing uncatalyzed and acid-catalysed pathways via DFT and DLPNO-CCSD(T) calculations. While the topic is potentially relevant to covalent adaptable networks (CANs), the study suffers from fundamental scientific shortcomings that render it unsuitable for publication in this journal, regardless of revision.

The most serious flaw lies in the disconnect between the computational treatment and the experimental reality of CAN materials. All calculations are performed at standard temperature (298.15 K), whereas the experimental systems operate between 50°C and 180°C. No meaningful attempt is made to model thermodynamic properties at elevated temperatures. The omission of temperature-dependent corrections to entropy and enthalpy, particularly in a context involving polymer melts and viscoelastic environments, undermines the validity of the reported Gibbs free energy barriers (44.1 and 25.9 kcal/mol). These values cannot be used to infer real-world kinetics or material behaviour and should not be interpreted as such.

The reliance on an extremely simplified model system—dimers of methacrylic and 4-pentenoic anhydride—further weakens the conclusions. The authors do not account for any of the physical constraints relevant in polymeric systems, such as chain entanglement, local dielectric heterogeneity, or stress-strain coupling. Despite this, they draw conclusions about CAN performance and recyclability, which are unwarranted extrapolations. The mechanistic discussion is also incoherent: an initial set of unrealistic, high-barrier mechanisms (including concerted 4- and 6-membered ring transitions) are presented before being discarded in favour of a more plausible stepwise mechanism. Including these discarded pathways in the core figures and schemes dilutes the scientific message and reveals a lack of focus.

The conformational sampling work, performed using CREST and DFT refinement, is a welcome addition. However, this makes the first part of the study redundant, as it is unclear why energy profiles are presented both with and without conformational analysis. Have the authors used Boltzmann-weighted free energy minima? Transition state barriers differ by small margins (~1–2 kcal/mol), making it essential to apply rigorous statistical thermodynamics. Importantly, the potential role of non-covalent interactions in shaping these barriers is not thoroughly explored.

A particularly troubling aspect is the inconsistent and poorly justified use of basis sets and computational protocols. The manuscript applies both Karlsruhe (def2) and Pople-style (6-31+G(d)) basis sets interchangeably for geometry optimisations and single-point corrections, without offering any rationale. This arbitrary mixing of def2 and Pople families across gas-phase and solvent-phase steps is methodologically unsound and reveals a lack of understanding of the differences in parametrisation, valence space resolution, and compatibility with modern DFT functionals. The simultaneous use of M06-2X/def2-SVP and M06-2X/6-31+G(d) geometries, followed by single points at M06-2X/def2-TZVP and M06-2X/6-31++G(d,p), respectively—then overlaid with SMD(Chloroform)—is not just inconsistent but chemically meaningless. Such inconsistencies render the computed energy surfaces difficult to interpret and impossible to reproduce in a principled way. This uncritical combination of Gaussian and ORCA calculations, basis sets of differing quality, and solvent models applied post hoc shows a fundamental lack of rigour in computational practice. A benchmark using different DFT functionals arbitrarily is nonsense.

The computational approach, while broadly acceptable in scope, is not executed to the standard expected in a leading computational chemistry journal. The M06-2X/def2-SVP level used for geometry optimisation lacks the flexibility needed for dispersion-rich or sterically congested transition states. No explicit dispersion corrections (e.g., D3) are included. While DLPNO-CCSD(T)/CBS single-point energies add some rigour, they are insufficient to compensate for these limitations, especially when dealing with charge-separated or strained transition states.

Mechanistically, the proposed acid-catalysed pathway lacks key details. The authors assume unspecified "traces of acid" can lower the reaction barrier but provide no information about protonation sites, counterions, or the stability of charged intermediates under melt-phase conditions. No kinetic modelling, proton exchange analysis, or solvent coordination study is performed. The barrier of 25.9 kcal/mol is in numerical agreement with one reported experimental value, but the mechanistic explanation offered is speculative and unsupported.

Additionally, the study is weakened by the absence of a supporting bonding analysis standard in the mechanistic work of this kind. There is no use of AIM (for bond critical points), NBO (for hyperconjugation or lone pair participation), or EDA (to quantify non-covalent interactions). These omissions leave the reaction pathways undercharacterised and prevent a rigorous description of the transition state electronic structure.

The Supporting Information is also insufficiently detailed: the absence of Cartesian coordinates and vibrational frequency data prevents reproducibility and external verification. The kinetic analysis via a four-point Arrhenius plot is statistically meaningless and should be removed.

In summary, this work is a superficial theoretical exercise that does not meet the scientific standards of this journal. It lacks the depth, rigour, and contextual relevance required for serious publication in a top-tier outlet. Even with substantial improvement, including proper temperature modelling, mechanistic refinement, bonding analysis, and coherent computational protocols, the work might only be suitable for a more modest journal focused on physical organic chemistry, such as *The Journal of Organic Chemistry*. However, as it stands, I do not recommend this manuscript for publication in this journal under any circumstances.

Reviewer #2

(Remarks to the Author)

This paper reports the analysis of anhydride exchange reactions via DFT methods. The reaction examined is between pentenoic anhydride and methacrylic anhydride, a system for which the experimental exchange reaction dynamics have been reported. Both uncatalyzed and acid-catalyzed reactions were studied. The authors have been thorough in examining/optimizing methodology, conformations, and reaction pathways. The data presented seems reasonable, and the conclusions are sound. I recommend it for publication after a few minor details are addressed (see below).

1. Did the authors consider any interactions between the pi bonds of the carbonyl and the ene groups (in either compound; e.g., carbonyl of pentenoic anhydride with the C=C bond in the methacrylic anhydride, and vice versa) in the various conformers/transition states? It would seem that the C=C group in the pentenoic acid, in particular, has enough flexibility such that it could wrap back over the C=O bond of the anhydride groups.
2. Reference 20 (Tillman et al.) has some kinetic data showing that acid-catalyzed exchange is indeed faster than uncatalyzed (3x faster at 35C with 5% acid based on the data in the SI of that paper). Is it possible that the calculations of the present manuscript can predict this difference?

Reviewer #3

(Remarks to the Author)

This study presents a comprehensive investigation into the mechanism of covalent bond exchange in anhydride-based Covalent Adaptable Networks (CANs) through the application of Density Functional Theory (DFT) calculations. The research specifically examines two anhydride monomers, methacrylic anhydride and 4-pentenoic anhydride, which are of significant interest due to their incorporation in vitrimers—a class of polymers characterized by dynamic covalent bonds that facilitate reprocessing, recycling, and self-healing properties.

The computational analysis proposes a detailed, stepwise bond exchange mechanism. In this mechanism, the cleavage of a C–O bond in one anhydride unit initiates an attack on the carbonyl carbon (C=O) of a second anhydride. This process leads to the formation of a quaternary carbon intermediate. Subsequently, a six-membered ring transition state promotes acyl migration, ultimately resulting in the formation of a mixed anhydride product.

Two distinct reaction pathways are explored extensively: the uncatalyzed mechanism and an acid-catalyzed mechanism. The study provides a thorough energetic and mechanistic comparison of both pathways, complemented by a correlation analysis with existing experimental data. This combined computational and experimental approach lends robustness to the proposed mechanism.

The authors conclude that the acid-catalyzed pathway plays a pivotal role in the dynamic bond exchange process, particularly in terms of enabling the reprocessability of anhydride-based CANs. This finding underscores the critical influence of residual acid species in facilitating the covalent adaptability of these materials.

The computational methods employed are rigorous and appropriate, and the results are presented in a clear and concise manner. The analysis effectively supports the conclusions, providing valuable insights into the reaction mechanisms to understand the bond exchange in anhydride-based CANs. Given the strength of the methodology and the clarity of the findings, I believe the article is suitable for publication in its current form.

Version 1:

Reviewer comments:

Reviewer #4

(Remarks to the Author)

I co-reviewed this manuscript with one of the reviewers who provided the listed reports. This is part of the Communications Chemistry initiative to facilitate training in peer review and to provide appropriate recognition for Early Career Researchers who co-review manuscripts.

Reviewer #5

(Remarks to the Author)

Zhang and co-workers present a detailed computational study showing how acid could catalyze anhydride bond exchange in CANs, offering insights on how to better design and control these dynamic cross-linkers. Although a simplified model was adopted, conformational searches and the consideration of various effects were incorporated. The key contribution is that the authors highlight the influence of the acid, which even in residual/trace amounts, could play a substantial role in catalyzing bond metathesis. Such an effect has often been overlooked but is now receiving increased attention. Overall, the manuscript is suitable for publication after addressing the minor issues below:

1. In the TOC graphic, label the substituents as R1 and R2 to underscore the method's generality and its applicability for forming unsymmetrical anhydrides. While this may be less critical for CANs, it could be relevant to other applications.
2. In the graphical TOC, a double dagger(‡) highlights transition states (TS) to emphasize bond forming/breaking. Please apply the same convention in Scheme 1 panels ii) and iii) by explicitly indicating which bonds are changing in the TS. If that is not feasible, remove the ‡ and present these species as intermediates instead.
3. The summary of the DFT benchmarking approach and results (p. 7–8) should be revised: An explanation of how RMSD values were obtained should be provided, "Gibbs energies" should be changed to "Gibbs free energies" in multiple places, and a clearer overview of how benchmarking was carried out should be provided, e.g. the paragraph should state more clearly how relative energies were obtained.
4. Throughout the manuscript Gibbs energy is used instead of Gibbs free energy.
5. Scheme 2 i)-a)—arrow-pushing: The curved arrow on the upper right appears to originate from an oxygen lone pair, which would lead only to an intermediate. In this case, the more appropriate depiction is an arrow originating from the relevant σ -bond to represent the concerted rearrangement. Overall, the origins of some of the arrows in Scheme 2 and the GTOC should be shown more clearly and for clarity, arrows should be drawn in a way that they don't cross over the chemical structure.
6. Scheme 2 i)-b): The double bond is difficult to discern (likely a formatting issue). Please improve the rendering/line weight or resolution.
7. Scheme 2 ii), If the starting geometry was drawn with blue anhydride at the bottom and the black on the top to maintain the consistency with the rest of the structure, this would make the mechanism easier to follow.
8. The choice of solvent, CHCl_3 should be connected to the experimental values that were measured in CDCl_3 , this should be mentioned as background information when describing the computational methods.
9. Especially since the computational methods are only described in detail in the Supporting Information, the precise level of theory, including solvent correction and, where relevant, temperature should be given in figure captions. E.g. instead of describing the structures as "DFT-optimised", the caption of Figure 2 should give the level of theory that was used so that a reader can appreciate accuracy and limitations.
10. For clarity, the computational methods used throughout the manuscript should be summarised in a dedicated section that is separate from the description of the benchmarking procedure. While the first paragraph of section 2.1 in the manuscript provides some information, a concise overview and rationale of the types of calculations performed (i.e. optimizations, frequency calculations, IRC, and single point calculations) would be useful.
11. The temperature used in the calculations as well as the use of change of state corrections from 1 atm to 1 M should be more clearly stated where computational methods are provided. Were change of state corrections calculated?
12. There are some formatting issues with " CDCl_3 .21" in SI II.9 Effect of Temperature on Gibbs energy correction.
13. The authors performed extensive additional calculations to support section 2.4 of the manuscript. It would be helpful to the reader if key results of the frontier molecular orbital and distortion-interaction analyses were shown through a figure in this manuscript section rather than through references to the SI.
14. Coote, Truong, Anastasaki and co-workers have previously shown that acid catalyzes the self-initiation of polymerization in vinyl monomers (Nat. Synth. 2024, 3 (3), 347–356.), which is closely related to the topic the authors present here and hence should be discussed. The authors could consider briefly discussing that in the real environment of a polymer matrix the entropic loss will be less than in the bimolecular model used (i.e. treated two reactants as separate molecules), meaning a lower reaction barrier can be expected.

Overall, our impression is that the authors took into consideration the critical comments of reviewer 1 and endeavoured to provide thorough responses.

The authors addressed the aspect of temperature, which reviewer 1 pointed out as the most critical flaw of the manuscript by recalculating thermochemical values at a range of temperatures, showing that their initial conclusions regarding rate determining steps and catalysis remain unchanged. In this context, it is worth pointing out that for example, the ideal gas

model with the harmonic oscillator also represents an approximation, but the results are still able to shed light on the possible mechanism and this isn't commonly criticised. While calculating thermochemical values at the experimental temperature is best practice, there are many examples of useful publications in the computational chemistry literature that present results calculated at room temperature and still provide valuable qualitative insights into experimental observations obtained at different temperatures. We don't see this shortcoming as critical enough to entirely reject all conclusions from the study and the authors have now addressed this aspect.

Both the model system, as well as the computational methods have shortcomings as there will always be a trade-off between accuracy of computational methods and the system size that can be studied within the constraints of available computational resources. The authors are correct in stating that the M062X DFT functional was parameterised with dispersion interactions in mind although some benchmarking studies do suggest that performance with regard to dispersion interactions can be improved through the addition of a dispersion correction. However, our impression is that the uncertainty introduced by approximations in the implicit solvent model (again, a common and accepted choice) could arguably be considered as larger than the error resulting from insufficient description of dispersion interactions. Shortcomings in the description of solute-solvent interactions reducing dispersion interactions in a real systems vs a computational models and shortcomings in the description of dispersion interactions can cancel, (fortuitously) leading to reasonable performance. The benchmarking and conformational searching carried out by the authors of this study should be sufficient to provide a qualitative picture of the importance of acid catalysis. Ideally, thorough conformational searches should be carried out on all structures at a higher level of theory and it has been shown that ordering of conformers can suffer when systematic conformational searches of all possible bond rotations are not carried out using a reasonable DFT method, but we appreciate that this can come at a substantial computational cost, especially in the case of "floppy" systems, such as those in the present study. While this is beyond the scope of the current study, we would recommend follow-up work using multi-scale modelling as well as molecular dynamics, such as QM/MM with free energy sampling, for further insights.

An aspect that the authors should consider addressing is the presentation of the computational methods as per our comments, as well as further elaboration on the analyses (i.e. NBO and activation-strain analyses) they carried out in response to recommendations by reviewer 1. A concise presentation of the computational methods separate from the description of the benchmarking procedure will make it easier for a reader to quickly gauge the accuracy and limitations that can be expected.

Lastly, as the authors correctly pointed out in their response letter, deposition of their computational data in the Zenodo research data repository and is arguably the more accessible and more modern way of ensuring reproducibility. While they could have opted to also present coordinates in the traditional format in the supporting information in addition to the reference to the Zenodo database, this is not an issue.

We hope that these additional comments with respect to the response to reviewer report 1 are helpful.

Point-by-Point Response

>> Reviewer #1:

Comment 1:

While the topic is potentially relevant to covalent adaptable networks (CANs), the study suffers from fundamental scientific shortcomings that render it unsuitable for publication in this journal, regardless of revision.

Response:

We thank the reviewer for his/her appreciation of our topic's potential relevance to covalent adaptable networks (CANs) and appreciate his/her concerns regarding the scientific shortcomings of our manuscript. We have endeavoured to carefully and thoroughly consider all the comments and perform additional studies to improve our manuscript, which benefits greatly from the critical yet constructive feedback that this reviewer has provided.

Comment 2:

“The most serious flaw lies in the disconnect between the computational treatment and the experimental reality of CAN materials. All calculations are performed at standard temperature (298.15 K), whereas the experimental systems operate between 50°C and 180°C. No meaningful attempt is made to model thermodynamic properties at elevated temperatures. The omission of temperature-dependent corrections to entropy and enthalpy, particularly in a context involving polymer melts and viscoelastic environments, undermines the validity of the reported Gibbs free energy barriers (44.1 and 25.9 kcal/mol). These values cannot be used to infer real-world kinetics or material behaviour and should not be interpreted as such.”

Response:

We thank the reviewer for raising this relevant point. We initially calculated the thermochemistry at standard temperature (298.15 K) to match the experimental values for the kinetics measured for the model system using NMR. We have since recalculated all thermochemistries at additional different temperatures (50°C, 100°C, 150°C, 180°C, 200°C). The results have been added to the Supporting Information section II.9. Notably, the rate determining step remains unchanged.

We have modified the abstract to read as the following:

“The bond exchange barrier increases from 44.1 kcal/mol at 25 °C to 52.8 kcal/mol at 200 °C for the uncatalysed route and from 25.9 kcal/mol to 33.0 kcal/mol over the same temperature range for acid-catalysed route.”

and the following in the manuscript:

“We found the bond exchange barrier for uncatalysed route is as high as 44.1–52.8 kcal/mol whereas the barrier for acid-catalysed route can be lowered to 25.9–33.0 kcal/mol, suggesting

CREATING GROWTH, ENHANCING LIVES

Mission | We advance science and develop innovative technology to further economic growth and improve lives.

Vision | A global leader in science, technology and open innovation.

T +65 6419 1111

that anhydride dynamic covalent bonds could be a good candidate for high temperature applications and reprocessing or recycling can be achieved with acid catalyst under reasonable reaction temperatures.” and

“The high Gibbs free energy barrier of ~ 36 kcal/mol at 200°C (SI section II.7), obtained for acid/anhydride bond exchange in CANs prepared from random copolymer of n-butyl acrylate and acrylic acid,²³ agrees well with the predicted barrier of acid-catalysed mechanism of 33.0 kcal/mol at 200°C (SI section II.9).”

Comment 3:

“The reliance on an extremely simplified model system—dimers of methacrylic and 4-pentenoic anhydride—further weakens the conclusions. The authors do not account for any of the physical constraints relevant in polymeric systems, such as chain entanglement, local dielectric heterogeneity, or stress-strain coupling. Despite this, they draw conclusions about CAN performance and recyclability, which are unwarranted extrapolations.

The mechanistic discussion is also incoherent: an initial set of unrealistic, high-barrier mechanisms (including concerted 4- and 6-membered ring transitions) are presented before being discarded in favour of a more plausible stepwise mechanism. Including these discarded pathways in the core figures and schemes dilutes the scientific message and reveals a lack of focus.”

Response:

We thank the reviewer for his/her concern. Our model systems aim to primarily elucidate the complex mechanism of the anhydride bond exchange, which has not been studied in detail, either experimentally or computationally, but has important applications in vitrimers and CAN reprocessing and recycling but yet to be studied theoretically.

We acknowledge this reviewer’s comment. To understand the role of constraints on model anhydride bond exchange mechanism, we have a follow up study using the system indicated as shown below:

CREATING GROWTH, ENHANCING LIVES

Mission | We advance science and develop innovative technology to further economic growth and improve lives.

Vision | A global leader in science, technology and open innovation.

T +65 6419 1111

In this follow up system, by varying the chain lengths of the model systems, we aim to study how the strains will contribute to the barrier changes in our proposed mechanism. We also plan to follow up this system with classical MD and QM/MM on realistic vitrimer models to understand the effect of polymer environment on the barrier heights.

Preliminary PES for this model is shown below, which concurs with our report that the first step is the overall rate-determining step and that this barrier can be quite high. However, the effect of polymer strain will be further modelled using systems of varying lengths and MD methods but this will be the topic of our next investigation, after our present work whose *main* focus is the detailed study of the anhydride exchange mechanism.

For the mechanistic discussion, we respectfully disagree that the discussion is incoherent, as we do not know *a priori* that the explored mechanisms are unrealistic and have high barriers. It is important to explore these possibilities (including concerted 4- and 6-membered ring transitions). We explore all possibilities of uncatalyzed reactions to show that this is very difficult (under normal conditions) to happen if there are no catalysts present. This is an important conclusion of the manuscript. It is not uncommon for computational mechanistic papers to discuss different possibilities and conclude which ones are less probable and which one more probable.

Comment 4:

“The conformational sampling work, performed using CREST and DFT refinement, is a welcome addition. However, this makes the first part of the study redundant, as it is unclear why energy profiles are presented both with and without conformational analysis. Have the authors used Boltzmann-weighted free energy minima? Transition state barriers differ by small margins (~1–2 kcal/mol), making it essential to apply rigorous statistical thermodynamics. Importantly, the potential role of non-covalent interactions in shaping these barriers is not thoroughly explored.”

CREATING GROWTH, ENHANCING LIVES

Mission | We advance science and develop innovative technology to further economic growth and improve lives.

Vision | A global leader in science, technology and open innovation.

T +65 6419 1111

Response:

We thank the reviewer for this comment. We reasoned that for the initial assessment and benchmarking, whichever step/pathway we use will not affect the benchmarking performance and results substantially, as long as we kept conformational orientations consistent. This is what many computational chemists do during modelling and simulation studies, without devoting much attention to conformational effects, while keeping relative conformations consistent throughout the reaction pathway. To avoid the confusion arising from including the PESs with and without conformational sampling, we have moved Figure 1 in the main text, where the results are based on the study of 4-m pathway without conformational sampling, to the supporting information, Figure S1. We have also added the following in the SI to clarify this:

“In studying this pathway, we kept all relative conformations and orientations of side groups consistent throughout. Note that although we later found a more favourable pathway via 6-membered TSs (Scheme 2ii, pathway a), for the purpose of initial benchmarking of DFT levels of theory, it is sufficient to consider any reaction pathway.”

For Boltzmann-weighted free energy profiles, we performed such studies using the conformers we located for the dynamic covalent bond exchange via 6-membered TSs via Pathway I and present this in the Supporting Information Section “**II.11 Effect of Boltzmann weighting in resulting Gibbs energy profile**”. We note that the energy profile, reproduced below, does not change substantially and indeed observe that the first step remains as the overall rate-determining step:

We have added the following discussion in the manuscript:

“We further note that with Boltzmann weighting (Section II.11), the PES does not change substantially and the first step remains as the rate-determining step.”

For non-covalent interactions, see response to comment 8.

CREATING GROWTH, ENHANCING LIVES

Mission | We advance science and develop innovative technology to further economic growth and improve lives.

Vision | A global leader in science, technology and open innovation.

T +65 6419 1111

Comment 5:

“A particularly troubling aspect is the inconsistent and poorly justified use of basis sets and computational protocols. The manuscript applies both Karlsruhe (def2) and Pople-style (6-31+G(d)) basis sets interchangeably for geometry optimisations and single-point corrections, without offering any rationale. This arbitrary mixing of def2 and Pople families across gas-phase and solvent-phase steps is methodologically unsound and reveals a lack of understanding of the differences in parametrisation, valence space resolution, and compatibility with modern DFT functionals. The simultaneous use of M06-2X/def2-SVP and M06-2X/6-31+G(d) geometries, followed by single points at M06-2X/def2-TZVP and M06-2X/6-311++G(d,p), respectively—then overlaid with SMD(Chloroform)—is not just inconsistent but chemically meaningless. Such inconsistencies render the computed energy surfaces difficult to interpret and impossible to reproduce in a principled way. This uncritical combination of Gaussian and ORCA calculations, basis sets of differing quality, and solvent models applied post hoc shows a fundamental lack of rigour in computational practice. A benchmark using different DFT functionals arbitrarily is nonsense.”

Response:

We would like to clarify that there is some misunderstanding by the reviewer – we did not use “Karlsruhe (def2) and Pople-style (6-31+G(d)) basis sets interchangeably for geometry optimisations”, in fact, these are different basis sets used for benchmarking and in no place is the energies from using different basis sets used in a single potential energy surface for comparison – in which case obvious errors would have occurred since mixing different basis sets will get the user very different absolute energy values. We have clearly specified these in the supporting information, where Karlsruhe or Pople basis sets were used by themselves in separate levels of theory for benchmarking:

“Four levels of theory were chosen for comparison: 1) SMD(CHCl₃)-M06-2X/def2-TZVP//M06-2X/def2-SVP; 2) SMD(CHCl₃)-M06-2X/def2-TZVP//SMD(CHCl₃)-M06-2X/def2-SVP; 3) SMD(CHCl₃)-M06-2X/6-311++G(d,p)//M06-2X/6-31+G(d) and 4) SMD(CHCl₃)-DLPNO-CCSD(T)-CBS(complete basis set, cc-pVDZ, cc-pVTZ)//M06-2X/def2-SVP.” We have clarified this further in the supporting information by including:

“we applied level of theory 4, SMD(CHCl₃)-DLPNO-CCSD(T)/CBS//M06-2X/def2-SVP, for all subsequent studies.”. We note that the labels for the level of theory in the energy profiles may contain typos and could have led to the wrong impression that basis sets were used interchangeably. We have removed these unnecessary labels and include in the captions where appropriate.

For benchmarking studies, it is not uncommon to have limited functional/basis set for benchmarking. In our case, the benchmarking is not the main focus of the study but to find a suitable combination of DFT functional M06-2X, basis set and single point correction methods. It should be noted that for practical purposes, authors usually pre-select functionals or basis sets, typically constrained by computational resources. Such selection of limited functional and/or basis sets for benchmarking, is of standard practice and can be seen across many studies,

CREATING GROWTH, ENHANCING LIVES

Mission | We advance science and develop innovative technology to further economic growth and improve lives.

Vision | A global leader in science, technology and open innovation.

T +65 6419 1111

even for ones purely benchmarking studies (*Chem. Phys. Lett.* **2025**, 870, 142065, *J. Phys. Chem. A* **2024**, 128, 4391, *J. Phys. Chem. A* **2023**, 127, 7943, *J. Chem. Theory and Comput.* **2023**, 19, 5036, *J. Phys. Chem. A*, **2021**, 125, 9838, *J. Chem. Theory Comput.* 2005, 1, 3, 415, *Int J Quantum Chem.* 2017;117:e25409, *Chem. Theory Comput.* 2023, 19, 22, 8365, *Phys. Chem. A* 2013, 117, 23, 4945, *Phys. Chem. Chem. Phys.*, 2025, **27**, 8706).

Comment 6:

“The computational approach, while broadly acceptable in scope, is not executed to the standard expected in a leading computational chemistry journal. The M06-2X/def2-SVP level used for geometry optimisation lacks the flexibility needed for dispersion-rich or sterically congested transition states. No explicit dispersion corrections (e.g., D3) are included. While DLPNO-CCSD(T)/CBS single-point energies add some rigour, they are insufficient to compensate for these limitations, especially when dealing with charge-separated or strained transition states.”

Response:

We thank the reviewer for this comment. M06-2X has already some empirical dispersion corrections added (*Theor. Chem. Acc.* **2008**, 120 (1), 215) and there are benchmarking studies showing that the addition of dispersion corrections (e.g., D3) in the Minnesota functionals such as M06-2X provides only marginal improvement, if at all. For example, in *Phys. Chem. Chem. Phys.*, 2017, **19**, 32184, it was stated that “Note that M06, MN15L, and MN15 are the only Minnesota DFAs that do not seem to benefit from the DFT-D3 correction.”; In *ChemPhysChem* 2011, 12, 3421 improvement for including D3 in M06-2X may be small). M06-2X has been benchmarked against many functionals and shown to perform very well for organic systems.

In fact, this paper shows that M06-2X with an UltraFine grid for Ar₂ converges smoothly whereas the M06-2X-D3 curve of Ar₂ shows small oscillations (*Int J Quantum Chem.* 2017; 117:e25358). In addition, M06-2X performs well even when benchmarked against dispersion corrected DFT functionals, for studying large organic molecules, e.g., in this study: Rayne, S., Forest, K. Performance of the M062X density functional against the ISOL set of benchmark isomerization energies for large organic molecules. *Nat Prec* (2010). <https://doi.org/10.1038/npre.2010.5183.1>. In addition, a benchmarking study (*J. Phys. Chem. A* 2013, 117, 47, 12590, which has been cited close to 700 times) using Minnesota functionals vs B3LYP suggests that M06 suite of functionals performs well for describing dispersion interactions:

“This general approach has been used in the M06 suite of functionals, which differ in the amount of the exact exchange included with M06 including 27% of the HF exchange, M06-2X including 54%, and M06-HF including 100%.^{23–25} The suite has proved highly successful at describing dispersion interactions for neutral molecular systems, particularly M06-2X, which has an s6 scaling factor of Grimme’s long-range dispersion correction of only 0.06.^{23,38”}

Nevertheless, to be rigorous, we took the lowest energy conformers of PES for the major pathway and performed additional studies by reoptimising the structures using M06-2X-D3,

CREATING GROWTH, ENHANCING LIVES

Mission | We advance science and develop innovative technology to further economic growth and improve lives.

Vision | A global leader in science, technology and open innovation.

T +65 6419 1111

followed by SP DLPNO-CCSD(T) with solvent correction. The following show the comparison of energy profiles, with the SMD(chloroform)-DLPNO-CCSD(T)//M06-2X-D3/def2-SVP results in square brackets and originally computed SMD(chloroform)-DLPNO-CCSD(T)//M06-2X/def2-SVP shown. We observed that the energy profile thus obtained is no different either with D3 correction or not, when using M06-2X functional, in agreement with other studies.

These additional results have now been included in section “II.10 Effect of inclusion of D3 correction in M06-2X in resulting Gibbs energy profile” of the supporting information.

Comment 7:

“Mechanistically, the proposed acid-catalysed pathway lacks key details. The authors assume unspecified "traces of acid" can lower the reaction barrier but provide no information about protonation sites, counterions, or the stability of charged intermediates under melt-phase conditions. No kinetic modelling, proton exchange analysis, or solvent coordination study is performed. The barrier of 25.9 kcal/mol is in numerical agreement with one reported experimental value, but the mechanistic explanation offered is speculative and unsupported.”

Response:

We thank the reviewer for his/her feedback. We acknowledge that the protonation site on the substrate should be more clearly specified. In terms of only using proton for protonation, we reasoned that this is acceptable, as shown in the study carried out by Stefan Grimme (*J. Comput. Chem.* **2017**, 38, 2618-2631) where his group showed that in a protonation site screening analysis based on the GFN-xTB method, it is sufficient to generate the protonated starting geometries, optimizing them, and ranking them by total energies.

In the study, the possible protonation centres are determined using Foster–Boys localised molecular orbitals (LMOs) to detect potential protonation sites such as lone pairs (LPs) and π orbitals at the tight-binding level, with each centre protonated to form a starting geometry optimised at a molecular charge of +1. A naked proton (H^+) is placed along the axis of the identified lone pair or at the centre of π orbitals to optimise the resulting protonated structures (protomers) and further refined using DFT for protomers within certain range of energy interval

CREATING GROWTH, ENHANCING LIVES

Mission | We advance science and develop innovative technology to further economic growth and improve lives.

Vision | A global leader in science, technology and open innovation.

T +65 6419 1111

and the results compared to identify the most viable protonation site for these organic and organometallic molecules with multiple possible protonation sites.

For our system (two anhydride monomers), there are only two possible protonation sites for each monomer, one on the O atom of the carbonyl group (likely linking/bridging those two O atoms of the two acyl groups), one on bridging O atom of the two acyl groups of the anhydride.

We performed additional studies to ascertain the relative Gibbs energy change in the protonation for all oxygen sites for the two monomers. We found that, as expected, in each monomer (MAA vs PNA), the protonation on the O atom of the carbonyl group is more favourable than the protonation on the bridging O atom. In fact, when we start from a guess structure by placing the proton on the bridging O atom, it optimises in such a way that the C–O bond of anhydride breaks and forms such that the bridging O atom becomes carbonyl O atom (see Gaussian log files *MAA_c1_H_on_anhydrideO_opt.log* and *PNA_c1_H_on_anhydrideO_opt.log* attached). In addition, the protonation of carbonyl O atom on MAA is more favoured over that on PNA, by 1.4 kcal/mol. We have added this in the supporting information, reproduced below:

“We follow the approach for studying the site of protonation by adding a naked proton, as reported by Grimme’s work.²⁵ Since our system has two distinct possible sites for protonation, namely the anhydride carbonyl O atom and the bridging O atom, we investigated the relative stabilities of the protonated species for each possibility for both MAA and PNA monomer. For both monomers, the starting structure where the proton is near the bridging O atom optimises to the structure where the proton is on one carbonyl O atom ([MAA-H+]_{O2} and [PNA-H+]_{O2}). The optimised DFT structures and their relative Gibbs energies are given in Figure S11.

[MAA-H+]_{O1} $\Delta\Delta G^\ddagger = 0.0 \text{ kcal mol}^{-1}$	[MAA-H+]_{O2} $\Delta\Delta G^\ddagger = 3.8 \text{ kcal mol}^{-1}$
	[PNA-H+]_{O1} $\Delta\Delta G^\ddagger = 1.4 \text{ kcal mol}^{-1}$	[PNA-H+]_{O2} $\Delta\Delta G^\ddagger = 3.6 \text{ kcal mol}^{-1}$

CREATING GROWTH, ENHANCING LIVES

Mission | We advance science and develop innovative technology to further economic growth and improve lives.

Vision | A global leader in science, technology and open innovation.

T +65 6419 1111

Figure S11. DFT optimised structures for the PNA and MAA monomer, with their Gibbs energies for the protonation process given relative to the most stable structure, [MAA-H+]₀₁, which forms the proton-bridged 6-member ring.

In addition, there are precedents of studies using naked proton source (H^+ or H_3O^+) in mechanistic studies (one example is *Angew. Chem. Int. Ed.* **2025**, e202500074, DOI: 10.1002/anie.202500074). Thus, we believe that this is an acceptable practice, where an unspecified source of proton may be used in the acid-catalysed mechanism.

Comment 8:

“Additionally, the study is weakened by the absence of a supporting bonding analysis standard in the mechanistic work of this kind. There is no use of AIM (for bond critical points), NBO (for hyperconjugation or lone pair participation), or EDA (to quantify non-covalent interactions). These omissions leave the reaction pathways undercharacterised and prevent a rigorous description of the transition state electronic structure.”

Response:

We thank the reviewer for this constructive feedback. We performed additional analyses to understand and characterise TSs and have added the results in the SI section “**II.12 Frontier molecular orbitals, natural bond orbital (NBO) charges, non-covalent interaction (NCI) plots and distortion-interaction/activation-strain analyses**”. In addition, we have added the relevant discussion in the manuscript, reproduced below:

“2.4 Electronic structure analyses of key TSs

To gain a better understanding of the electronic structure description of the transition states of both catalysed and uncatalysed pathways, we performed Natural Bond Orbital (NBO) charge analyses and frontier molecular orbital (FMO) calculations (HOMO/LUMO plots) for key transition states (see SI Section II.12). The NBO analyses revealed consistent electron distributions across all uncatalysed transition states (Figure S15): O atoms exhibit negative NBO charges ranging from -0.587 to -0.732 a.u., while the adjacent carbon atoms carry positive charges between 0.873 and 0.983 a.u. This supports a concerted mechanism involving nucleophilic attack of oxygen on electrophilic carbon. In contrast, in the acid-catalysed **TS1_{acid}**, one O atom shows a much less negative charge (-0.369 a.u., Figure S15), consistent with the proposed formation of an acylium ion-like intermediate ($R-C\equiv O^+$). Frontier orbital analysis further revealed that the HOMO is largely localized on peripheral π bonds ($C=C$), and the LUMO is distributed on carboxylate π^* orbitals (Figure S15). This may

CREATING GROWTH, ENHANCING LIVES

Mission | We advance science and develop innovative technology to further economic growth and improve lives.

Vision | A global leader in science, technology and open innovation.

T +65 6419 1111

suggest that the key electronic reactivity is driven by groups outside the 6-membered ring scaffold. In addition, activation-strain analysis^{58–60} indicates that the major barrier to bond exchange arises from distortion of the reactant fragments as they reorganise into the 6-membered TS structures, rather than favorable interaction energies (SI Table S6). The acid-catalysed TS may benefit from enhanced interaction stabilisation, possibly due to charge-dipole interactions, leading to much lower barriers.”

Comment 9:

“The Supporting Information is also insufficiently detailed: the absence of Cartesian coordinates and vibrational frequency data prevents reproducibility and external verification.

The kinetic analysis via a four-point Arrhenius plot is statistically meaningless and should be removed.”

Response: We thank the reviewer for this comment.

We would like to point out that all the DFT-optimised structures can be obtained freely of charge under open-access. For the DFT optimized structures, instead of giving the coordinates as pdf, which makes them hard to be used, if at all, we have uploaded all the optimized structures and absolute energies to a Zenodo repository (<https://zenodo.org/records/15488044>). This allows researchers interested in reproducing or building upon our work (e.g., using our TS structure as a start and modifying it to suit their systems) to easily access the data, promoting transparency and facilitating future studies without the need to extract coordinates from the PDF (which may be non-trivial, sometimes all the x-coordinates are extracted, followed by y-coordinates, followed by z-coordinates, instead of copying line by line, making the transfer of the coordinates to build the molecular system difficult. Additionally, sometimes the coordinates span different pages, making the reuse and examining of the structures difficult and time-consuming), such as shown below:

1	-0.14855000	1.63425100	1.36360800
6	-3.42921900	1.96560800	3.37520300
1	-3.35886000	-0.16507700	3.58386700
6	-2.82038300	3.10360900	2.86118200
1	-1.14097100	3.85923500	1.73580400
1	-4.36115000	2.04793900	3.93270300
1	-3.26993400	4.08364900	3.01315200
6	5.74691200	0.3071	-0.06384800
9	6.69628800	-0.56006400	0.29455900
9	6.04728200	0.75596800	-1.28904100
9	5.82755900	1.35922700	0.76135700
6	-4.01440300	-2.33563400	-0.87624500
6	-4.16741900	-2.38750400	-2.38880200
6	-2.95681100	-1.89516300	-3.16981100
6	-4.12045400	-0.92511800	-0.31807400
6	-2.48186400	-0.49297600	-2.99478500
6	-2.93164000	-0.02020900	-0.55264200
6	-2.42003400	0.30053500	-1.88869800
1	-4.79681800	-2.95625100	-0.41544200
1	-4.35962900	-3.42412900	-2.70057400

CREATING GROWTH, ENHANCING LIVES

Mission | We advance science and develop innovative technology to further economic growth and improve lives.

Vision | A global leader in science, technology and open innovation.

T +65 6419 1111

In fact, all our structures are readily available in .xyz format in open-access, citable and downloadable from the given link and can be visualized directly using visualization software such as Avogadro and PyMOL. We believe that this should be the way forward for efficient storage, reporting and reuse of DFT optimized structures, instead of having the coordinates printed in multiples of pages in a pdf file that is not directly accessible. Another advantage is that the deposition of these structures in open-access databases allow future researchers who would like to automate and access these structures for e.g., machine learning/AI in chemistry studies to easily automate and access these coordinates and structures without scraping through pages after pages for sections of cartesian coordinates in a pdf file.

This has already been specified in the SI:

“III. DFT-optimised structures and absolute energies

Geometries of all optimised structures (in .xyz format with their associated energy in Hartrees) are included in a separate folder named *DFT_structures_and_IRC_movie* with an associated readme.txt file. All these data have been uploaded to <https://zenodo.org/records/15488044> (DOI: 10.5281/zenodo.15488044).”

as well as the manuscript, in the initial submission:

“Data availability

DFT-optimised structures from this study in .xyz format have been uploaded to and are freely available at <https://zenodo.org/records/15488044> (DOI: 10.5281/zenodo.15488044).”

With all due respect, we quite strongly disagree that “the kinetic analysis via a four-point Arrhenius plot is statistically meaningless and should be removed”. All the data are from experimental measurements and this is the *de facto* method in almost if not all experimental studies of kinetics measurements. The two experimental studies from which we obtained the kinetic parameters are indeed from *two independent research groups* (*Polym. Chem.* **2020**, *11* (47), 7551 by Shipp et. al and *Cell Reports Phys. Sci.* **2021**, *2* (7), 100483 by Chen et. al). The reviewer may wish to clarify his/her reasoning on how they derive the conclusion that it is “statistically” meaningless and how many data points would be needed to be “statistically” significant, so that we can seek further ways to improve the manuscript constructively in this regard.

>> Reviewer #2:

Comment 1:

“The authors have been thorough in examining/optimizing methodology, conformations, and reaction pathways. The data presented seems reasonable, and the conclusions are sound. I recommend it for publication after a few minor details are addressed (see below).”

Response:

We thank the reviewer for his/her positive evaluation of our manuscript.

CREATING GROWTH, ENHANCING LIVES

Mission | We advance science and develop innovative technology to further economic growth and improve lives.

Vision | A global leader in science, technology and open innovation.

T +65 6419 1111

Comment 2:

“Did the authors consider any interactions between the pi bonds of the carbonyl and the ene groups (in either compound; e.g., carbonyl of pentenoic anhydride with the C=C bond in the methacrylic anhydride, and vice versa) in the various conformers/transition states? It would seem that the C=C group in the pentenoic acid, in particular, has enough flexibility such that it could wrap back over the C=O bond of the anhydride groups.”

Response:

We thank the reviewer for his/her comment on the flexibility of the side chain in forming interactions via the pi bonds. We have first considered the 6-membered ring transition state where either the acyl group/carbonyl group of the pentenoic anhydride or the methacrylic anhydride participate in the migration. Through conformational sampling, we allowed the molecules to adapt different conformations where the interactions are optimised at xTB level and then the lowest 20 xTB energy structures were further optimised at DFT level to identify the most stable complex/TS. We do observe that the side chain of the anhydride, particularly of pentenoic anhydride, can have some degree of flexibility, as the reviewer correctly hypothesised. However, due to the highly organised 6-membered ring TS, the orientations of the side chains are somewhat limited in the TSs. For example, we may observe that **TS1** and **TS1'** have the ene group of pentenoic acid “wrapped back” toward the 6-membered ring “core region” and interacting with other side groups but this is limited by the overall carbonyl group it is attached to that participates in the TS organisation.

Comment 3:

“Reference 20 (Tillman et al.) has some kinetic data showing that acid-catalyzed exchange is indeed faster than uncatalyzed (3x faster at 35C with 5% acid based on the data in the SI of that paper). Is it possible that the calculations of the present manuscript can predict this difference?”

Response:

We thank the reviewer for his/her very insightful observation! We calculated this using the activation barrier difference between the barrier of 44.1 kcal/mol for the uncatalyzed reaction and the barrier of 25.9 kcal/mol for the catalysed reaction ($\Delta\Delta G^\ddagger = 18.2$ kcal/mol), which translates to roughly 10^{13} to 10^{14} . This estimate, however, does not take into account the concentration or amount of the acid used. We have some good degree of confidence of our proposed reaction mechanism, however, as it currently stands, it seems that more factors need to be taken into account to be able to predict the rate differences between uncatalysed and acid-catalysed pathways.

CREATING GROWTH, ENHANCING LIVES

Mission | We advance science and develop innovative technology to further economic growth and improve lives.

Vision | A global leader in science, technology and open innovation.

T +65 6419 1111

>> **Reviewer #3:**

Comment 1:

“Two distinct reaction pathways are explored extensively: the uncatalyzed mechanism and an acid-catalyzed mechanism. The study provides a thorough energetic and mechanistic comparison of both pathways, complemented by a correlation analysis with existing experimental data. This combined computational and experimental approach lends robustness to the proposed mechanism.”

Response:

We thank the reviewer for his/her positive assessment of our work.

Comment 2:

“The authors conclude that the acid-catalyzed pathway plays a pivotal role in the dynamic bond exchange process, particularly in terms of enabling the reprocessability of anhydride-based CANs. This finding underscores the critical influence of residual acid species in facilitating the covalent adaptability of these materials.”

Response:

We thank the reviewer for his/her observation of the critical role of residual acids in covalent adaptability phenomenon, which can have applications in polymer recycling and reprocessing.

Comment 3:

“The computational methods employed are rigorous and appropriate, and the results are presented in a clear and concise manner. The analysis effectively supports the conclusions, providing valuable insights into the reaction mechanisms to understand the bond exchange in anhydride-based CANs. Given the strength of the methodology and the clarity of the findings, I believe the article is suitable for publication in its current form.”

Response:

We thank the reviewer for his/her positive and encouraging comments!

CREATING GROWTH, ENHANCING LIVES

Mission | We advance science and develop innovative technology to further economic growth and improve lives.

Vision | A global leader in science, technology and open innovation.

T +65 6419 1111

Point-by-Point Response

>> Reviewer #4:

Reviewer #4 co-reviewed with one of his/her junior colleagues/ECRs and did not give specific comments. We thank Reviewer #4 for his/her time in guiding the junior colleague in reviewing our work.

>> Reviewer #5:

Comment 1:

“Overall, the manuscript is suitable for publication after addressing the minor issues below:”

Response:

We thank the reviewer for his/her positive evaluation of our manuscript and support for our work’s publication at Communications Chemistry after minor revisions.

Comment 2:

“In the TOC graphic, label the substituents as R1 and R2 to underscore the method’s generality and its applicability for forming unsymmetrical anhydrides. While this may be less critical for CANs, it could be relevant to other applications.”

Response:

We thank the reviewer for his/her observation of the implicated mechanistic generality. We have revised the TOC graphic accordingly to reflect this.

Comment 3:

“In the graphical TOC, a double dagger(‡) highlights transition states (TS) to emphasize bond forming/breaking. Please apply the same convention in Scheme 1 panels ii) and iii) by explicitly indicating which bonds are changing in the TS. If that is not feasible, remove the ‡ and present these species as intermediates instead.”

Response:

We thank this reviewer for his/her acute observation and detailed checking. We have removed the double dagger (‡) to show that these species are intermediates through which the reactions pass in Scheme 1 panels ii) and iii).

CREATING GROWTH, ENHANCING LIVES

Mission | We advance science and develop innovative technology to further economic growth and improve lives.

Vision | A global leader in science, technology and open innovation.

T +65 6419 1111

Comment 4:

“The summary of the DFT benchmarking approach and results (p. 7–8) should be revised: An explanation of how RMSD values were obtained should be provided, “Gibbs energies” should be changed to “Gibbs free energies” in multiple places, and a clearer overview of how benchmarking was carried out should be provided, e.g. the paragraph should state more clearly how relative energies were obtained.”

Response:

We thank the reviewer for his/her insightful suggestions. We have added the following discussion in the SI to explain how RMSD values were obtained:

“The RMSD values between structures under alignment were obtained by applying the align function in PyMOL software, which first uses a global dynamic-programming (Needleman–Wunsch²³) alignment with BLOSUM62 scoring²⁴ to match atoms between the two structures. All atoms are superimposed by applying a least-squares fitting using the Kabsch algorithm²⁵ to minimize RMSD value. After this initial alignment, PyMOL performs up to five cycles of iterative refinement, discarding atoms with large deviations (typically beyond ~ 2 Å or two standard deviations) and repeating the fit until convergence. The final RMSD value after these procedures is then reported.”

All mentions of “Gibbs energies” have been changed to “Gibbs free energies” in both the manuscript and the SI.

For the details on the relative energies, we have added the following description in the SI: “Within each level of theory, the Gibbs free energy for each species is taken relative to the sum of the Gibbs free energies of the reactants (MAA and PNA) (Figure S1).”

Comment 5:

“Throughout the manuscript Gibbs energy is used instead of Gibbs free energy.”

Response:

We thank the reviewer for making this observation; we have changed all mentions of “Gibbs energies” to “Gibbs free energies”.

Comment 6:

“Scheme2 i)-a)—arrow-pushing: The curved arrow on the upper right appears to originate from an oxygen lone pair, which would lead only to an intermediate. In this case, the more appropriate depiction is an arrow originating from the relevant σ -bond to represent the concerted rearrangement. Overall, the origins of some of the arrows in Scheme 2 and the GTOC should be shown more clearly and for clarity, arrows should be drawn in a way that they don’t cross over the chemical structure.”

CREATING GROWTH, ENHANCING LIVES

Mission | We advance science and develop innovative technology to further economic growth and improve lives.

Vision | A global leader in science, technology and open innovation.

T +65 6419 1111

Response:

We thank the reviewer for his/her comments. We have changed/modified the arrows in the GTOC, Scheme 2 and Figure 4 to conform to the useful suggestions made by this reviewer – now, no arrows cross over the chemical structures.

Comment 7:

“Scheme 2 i)-b): The double bond is difficult to discern (likely a formatting issue). Please improve the rendering/line weight or resolution.”

Response:

We thank the reviewer for his/her comment on this. We tried to make the double bonds in Scheme 2 i)-b) all visible and consistent with all other double bonds in the manuscript.

Comment 8:

“Scheme 2 ii), If the starting geometry was drawn with blue anhydride at the bottom and the black on the top to maintain the consistency with the rest of the structure, this would make the mechanism easier to follow.”

Response:

We thank the reviewer for his/her comment. In Scheme 2 ii), pathway going up, we have placed the blue anhydride at the bottom, to maintain consistency with the rest of the scheme.

Comment 9:

“The choice of solvent, CHCl₃ should be connected to the experimental values that were measured in CDCl₃, this should be mentioned as background information when describing the computational methods.”

Response:

We thank the reviewer for his/her useful suggestion. We have included this clarification in the newly added Computational Methods section in the manuscript:

“Solvent effect for chloroform was modelled using SMD implicit solvation to account for the effect of chloroform solvent that was used in the experimental measurement of anhydride exchange rates²⁰ using ¹³C NMR in CDCl₃.”

Comment 10:

“Especially since the computational methods are only described in detail in the Supporting Information, the precise level of theory, including solvent correction and, where relevant, temperature should be given in figure captions. E.g. instead of describing the structures as “DFT-optimised”, the caption of Figure 2 should give the level of theory that was used so that a reader can appreciate accuracy and limitations.”

CREATING GROWTH, ENHANCING LIVES

Mission | We advance science and develop innovative technology to further economic growth and improve lives.

Vision | A global leader in science, technology and open innovation.

T +65 6419 1111

Response:

We thank the reviewer for his/her suggestion. We have added the level of theory and temperature at which the values are reported in the captions of Figures 1, 2 and 3.

Comment 11:

“For clarity, the computational methods used throughout the manuscript should be summarised in a dedicated section that is separate from the description of the benchmarking procedure. While the first paragraph of section 2.1 in the manuscript provides some information, a concise overview and rationale of the types of calculations performed (i.e. optimizations, frequency calculations, IRC, and single point calculations) would be useful.”

Response:

We thank the reviewer for his/her useful suggestion. We have added a new section named “2. Computational Methods” in the revised manuscript.

Comment 12:

“The temperature used in the calculations as well as the use of change of state corrections from 1 atm to 1 M should be more clearly stated where computational methods are provided. Were change of state corrections calculated?”

Response:

We thank the reviewer for his/her suggestion. In the newly added section “2. Computational Methods”, we mentioned that the change of state corrections have been applied: “The free energies reported in Gaussian from gas-phase optimisations were further corrected using standard concentration of 1 mol/L,³⁷⁻⁴⁰ which were used in solvation calculations, instead of the gas-phase 1atm used by default in Gaussian program.”

Comment 13:

“There are some formatting issues with “CDCI3.21” in SI II.9 Effect of Temperature on Gibbs energy correction.”

Response:

We thank the reviewer for his/her meticulous checking of our manuscript and SI. It seems that the citation was not rendered properly as a superscript. We have fixed this.

Comment 14:

“The authors performed extensive additional calculations to support section 2.4 of the manuscript. It would be helpful to the reader if key results of the frontier molecular orbital and distortion-interaction analyses were shown through a figure in this manuscript section rather than through references to the SI.”

CREATING GROWTH, ENHANCING LIVES

Mission | We advance science and develop innovative technology to further economic growth and improve lives.

Vision | A global leader in science, technology and open innovation.

T +65 6419 1111

Response:

We thank the reviewer for his/her useful suggestion, we have added Figure 4 in the revised manuscript to show the NBO charges and FMO plots for both uncatalysed (TS1) and acid-catalysed (TS1_acid) reaction pathways.

Comment 15:

“Coote, Truong, Anastasaki and co-workers have previously shown that acid catalyzes the self-initiation of polymerization in vinyl monomers (*Nat. Synth.* 2024, 3 (3), 347–356.), which is closely related to the topic the authors present here and hence should be discussed. The authors could consider briefly discussing that in the real environment of a polymer matrix the entropic loss will be less than in the bimolecular model used (i.e. treated two reactants as separate molecules), meaning a lower reaction barrier can be expected.”

Response:

We thank the reviewer for bringing our attention to the important contribution by Coote, Truong, Anastasaki and co-workers on acid-triggered self-initiation of vinyl monomers (*Nat. Synth.* 2024, 3, 347–356). This is highly relevant to our system, where protonation similarly plays a central role in generating radical species under mild conditions. We have now cited and discussed this work in the revised manuscript:

“The importance of acids in initialising and/or speeding up catalytic transformations has also been demonstrated in a study where Brønsted acids such as H₂SO₄ can catalyse the self-initiation of polymerisation in vinyl monomers, in the absence of external initiators.⁶⁹”

In addition, we acknowledge the reviewer’s insightful point on entropy effects and agree that the simplified model may not have captured the entropic changes effectively. We have clarified this limitation in the text:

“We note that our simplified mechanistic modelling treated the two reactants as independent molecules, which may overestimate the entropic penalty for bond formation. In the condensed polymer matrix, translational and rotational degrees of freedom of reacting groups are already restricted, thus reducing the effective entropic loss upon bond formation, and the actual free-energy barrier for such processes may be lower than predicted in our model.”

Comment 16:

“Overall, our impression is that the authors took into consideration the critical comments of reviewer 1 and endeavoured to provide thorough responses.”

Response:

We thank the reviewer for his/her positive and encouraging comments!

Comment 17:

“... , it is worth pointing out that for example, the ideal gas model with the harmonic oscillator also represents an approximation, but the results are still able to shed light on the possible mechanism and this isn’t commonly criticised.”

Response:

We thank the reviewer for his/her suggestion, we have cited some studies under the computational methods section, which references to computational works that were performed in similar fashion to yield good agreements with experimental results. We have also added the following line in the conclusion, to alert to the readers that our model is simplified, yet useful: “Despite a simplified model, our mechanistic elucidations provide valuable insights into the critical role of acids in otherwise challenging anhydride bond exchange processes.”

Comment 18:

“While this is beyond the scope of the current study, we would recommend follow-up work using multi-scale modelling as well as molecular dynamics, such as QM/MM with free energy sampling, for further insights.”

Response:

We thank the reviewer for his/her suggestion. We have added this in the last sentence of the conclusion:

“Future work will ... using multi-scale modelling as well as molecular dynamics.”

Comment 19:

“An aspect that the authors should consider addressing is the presentation of the computational methods as per our comments, as well as further elaboration on the analyses (i.e. NBO and activation-strain analyses) they carried out in response to recommendations by reviewer 1. A concise presentation of the computational methods separate from the description of the benchmarking procedure will make it easier for a reader to quickly gauge the accuracy and limitations that can be expected.”

Response:

We thank the reviewer for his/her useful suggestions and have added the section on “2. Computational Methods” and added the NBO and FMO plots in Figure 4.

Comment 20:

“Lastly, as the authors correctly pointed out in their response letter, deposition of their computational data in the Zenodo research data repository and is arguably the more accessible and more modern way of ensuring reproducibility. While they could have opted to also present coordinates in the traditional format in the supporting information in addition to the reference to the Zenodo database, this is not an issue.”

Response:

We thank the reviewer for his/her understanding of our preferred way of data reporting and would like to re-iterate that this data deposition under open-access will be beneficial in the long run for the computational chemistry community, as researchers can easily obtain, verify and reuse these data more efficiently.

Last but not least, we would like to specifically thank this reviewer (on top of all other reviewers), who have spent time and effort to check our work, both the manuscript and the supporting information, very carefully and meticulously and providing insightful and constructive comments for our work’s revision – the quality of our work has benefited greatly from these comments!

CREATING GROWTH, ENHANCING LIVES

Mission | We advance science and develop innovative technology to further economic growth and improve lives.

Vision | A global leader in science, technology and open innovation.

T +65 6419 1111